# Localised Natural Causal Learning Algorithms for Weak Consistency Conditions

**Kai Z. Teh**[1]  **Kayvan Sadeghi**[1]  **Terry Soo**[1]

[1]Department of Statistical Science, UCL, London, UK

## Abstract

By relaxing conditions for "natural" structure learning algorithms, a family of constraint-based algorithms containing all exact structure learning algorithms under the faithfulness assumption, we define localised natural structure learning algorithms (LoNS). We also provide a set of necessary and sufficient assumptions for consistency of LoNS, which can be thought of as a strict relaxation of the restricted faithfulness assumption. We provide a practical LoNS algorithm that runs in exponential time, which is then compared with related existing structure learning algorithms, namely PC/SGS and the relatively recent Sparsest Permutation algorithm. Simulation studies are also provided.

## 1 INTRODUCTION

Inferring causal relationships has always been of great interest in different fields, with some frameworks like potential outcomes and graphs gaining prominence amongst the causality community. A main goal of graph-based causal inference is causal discovery; given data, we would like to uncover the underlying causal structure in the form of a true causal graph, on which conventional graph-based causal inference techniques hinge. We will mostly be concerned with the setting of observational data only, such as when interventional data in the form of randomised control trials are unavailable, from which the true causal graph is recoverable up to its graphical separations. Current causal discovery approaches can generally be categorised into score-based approaches [Chickering, 2002] and constraint-based approaches [Spirtes et al., 2000]. Here, we will mostly be concerned with the latter.

Assumptions are needed for constraint-based approaches, otherwise many causal structures representing the same data may be obtained, resulting in vacuous causal statements.

Amongst these, the most common and widely known is the faithfulness assumption, where every conditional independence in the data generating distribution is exactly represented by the true causal graph [Zhang and Spirtes, 2008]; most constraint-based learning approaches such as PC and SGS provably return the true causal graph up to its graphical separations. However in practice and theory, the condition can be too strong at times [Uhler et al., 2013].

Efforts to relax the faithfulness assumption include the Sparsest Permutation (SP) algorithm by Raskutti and Uhler [2018], which provably returns the graphical separations of the true causal graph under strictly weaker assumption than faithfulness, at the expense of factorial run time by permuting the causal variables. Greedy approaches to speed up the SP algorithm [Solus et al., 2021, Lam et al., 2022] have been proposed, however these algorithms only return the true causal graph under strictly *stronger* conditions than SP.

Addressing this, Sadeghi and Soo [2022] proposed the class of "natural" structure learning algorithms, which under the faithfulness assumption, encompasses constraint-based approaches such as SGS/PC algorithms. In addition, natural structure learning algorithms are also proven to return the true causal graph up to graphical separations under well defined assumptions that are shown to be strictly weaker than faithfulness. Thus, the objective of this paper is as follows: 1.) To further weaken the consistency conditions by defining a localised version of natural structure learning algorithms. 2.) To provide a practical algorithm of this type that works under these conditions.

The structure of the paper will be as follows: Section 2 covers the relevant background, Section 3 covers the theory and practical algorithm and Section 4 compares the algorithm with related existing algorithms, PC and SP. All proofs will be deferred to the supplementary material.

## 2 BACKGROUND

### 2.1 GRAPHS

We first introduce the relevant concepts in graphical models, as well as some existing results in literature. In this work, unless noted otherwise, graphs will be implicitly assumed to be a *directed acyclic graph* (DAG) that is a graph over the set of nodes $V = \{1, ..., n\}$, with directed edges such that there does not exist a sequence of directed edges from a node to itself. We denote $\text{an}_G(C)$ as the set of nodes $i \notin C$ such that there exists a sequence of directed edges from $i$ to some $j \in C$ in $G$.

We denote $A \perp_G B \,|\, C$ as graphical separation in graph $G$ between $A, B$ given $C$, where $A, B, C \subseteq V$ are disjoint; in the case of DAGs, this can be understood as d-separation. A set of random variables $X = (X_1, ..., X_n)$ with joint distribution $P$ is associated to the set of nodes $V$. We denote $A \perp\!\!\!\perp B \,|\, C$ as conditional independence of $(X_i)_{i \in A}$ and $(X_j)_{j \in B}$ given $(X_k)_{k \in C}$. We relate the two notions together using *Markov property*:

**Definition 1** (Markov property). *A distribution $P$ is* Markovian *to $G$ if $A \perp_G B \,|\, C \Rightarrow A \perp\!\!\!\perp B \,|\, C$ for all disjoint $A, B, C \subseteq V$.*

If we have the reverse implication as well, then we have faithfulness:

**Definition 2** (Faithfulness). *A distribution $P$ is* faithful *to $G$ if $A \perp_G B \,|\, C \iff A \perp\!\!\!\perp B \,|\, C$ for all disjoint $A, B, C \subseteq V$.*

Sadeghi [2017] has shown that $P$ being faithful to DAG $G$ implies that $P$ satisfies *ordered upward stability* and *ordered downward stability* wrt $G$, defined in the case of DAGs, as follows:

**Definition 3.**

1. (Ordered upward stability (OUS)). *$P$ satisfies ordered upward stability wrt $G$ if for all $i, j, k$ and $C \subseteq V \setminus \{i, j, k\}$, such that $k \in \text{an}_G(i, j)$, we have $i \perp\!\!\!\perp j \,|\, C \Rightarrow i \perp\!\!\!\perp j \,|\, C \cup \{k\}$.*

2. (Ordered downward stability (ODS)). *$P$ satisfies ordered downward stability wrt $G$ if for all $i, j, k$ and $C \subseteq V \setminus \{i, j, k\}$, such that $k \notin \text{an}_G(i, j, C)$, we have $i \perp\!\!\!\perp j \,|\, C \cup \{k\} \Rightarrow i \perp\!\!\!\perp j \,|\, C$.*

If $P$ is faithful to $G$, then from $P$ we can recover the true causal graph $G$ up to its *Markov equivalence class (MEC)*, defined as the set of all graphs that imply the same graphical separations.

Denote $\text{sk}(G)$ to be the *skeleton* of graph $G$, formed by removing all arrowheads from edges in $G$. A *v-configuration*

is a set of nodes $i, k, j$ such that $i$ and $j$ are connected to $k$, but $i$ and $j$ are not connected, and will be represented as $i \sim k \sim j$. A v-configuration oriented as $i \rightarrow k \leftarrow j$ is a *collider*, otherwise the v-configuration is a *non-collider*.

**Remark 1.** *Some authors allow nodes $i$ and $j$ of collider $i \rightarrow k \leftarrow j$ to be adjacent, but we do not. If $i$ is not adjacent to $j$, then $i \rightarrow k \leftarrow j$ is sometimes called an unshielded collider, but will simply be referred to as a collider here.*

To relate $\text{sk}(G)$ with distribution $P$, we define $\text{sk}(P)$ as follows:

**Definition 4** ($\text{sk}(P)$). *Given a distribution $P$, the skeleton $\text{sk}(P)$ is the undirected graph with node set $V$, such that for all $i, j \in V$, the node $i$ is adjacent to $j$ if and only if there does not exist any $C \subset V \setminus \{i, j\}$ such that $i \perp\!\!\!\perp j \,|\, C$.*

Note that $\text{sk}(P)$ is the output of the skeleton building step of SGS/PC algorithm under faithfulness.

**Definition 5** (Adjacency faithfulness). *A distribution $P$ is adjacency faithful wrt graph $G$, if for all $i, j \in V$, we have: $i$ adjacent to $j$ in $G \Rightarrow i \not\perp\!\!\!\perp j \,|\, C$ for all $C \subseteq V \setminus \{i, j\}$.*

Note that if $P$ is adjacency faithful wrt $G$, then $\text{sk}(P) = \text{sk}(G)$.

### 2.2 NATURAL STRUCTURE LEARNING ALGORITHMS

Let $P$ be Markovian to the true causal graph $G_0$, then a causal learning algorithm aims to recover the graph $G_0$, up to the MEC, in which case we say that the algorithm is *consistent*. To relax the faithfulness assumption, Sadeghi and Soo [2022] introduced *natural structure learning algorithms*.

**Definition 6** (Natural structure learning algorithm). *An algorithm that takes distribution $P$ as input and outputs DAG $G(P)$ is natural if:*

1. $\text{sk}(G(P)) = \text{sk}(P)$.

2. *$P$ satisfies OUS and ODS wrt $G(P)$.*

The following conditions on $P$ and the true causal DAG $G_0$ ensure the consistency of natural structure learning algorithms:

**Definition 7** (V-stability). *$P$ is V-stable if for all v-configurations $i \sim k \sim j$ in $\text{sk}(P)$, and $C \subseteq V \setminus \{i, j, k\}$, the independencies $i \perp\!\!\!\perp j \,|\, C$ and $i \perp\!\!\!\perp j \,|\, C \cup \{k\}$ cannot both hold.*

**Remark 2.** *This is a definition on $P$ itself, and is implied by the well-known singleton transitive axiom, under adjacency faithfulness.*

**Proposition 1** (Theorems 14 and 25 of Sadeghi and Soo [2022]). *The graphs $G(P)$ and $G_0$ are Markov equivalent if the following holds:*

1. *$P$ satisfies adjacency faithfulness wrt $G_0$.*

2. *$P$ satisfies ordered upward and downward stabilities wrt $G_0$.*

3. *$P$ is V-stable.*

**Remark 3.** *In Sadeghi and Soo [2022], Condition 1 above is given in terms of converse pairwise Markovian instead of adjacency faithfulness, this is due to attempts in characterising the consistency conditions in terms of structural equation models (SEM). However, only the weaker adjacency faithfulness is needed and here we are focused on relaxing conditions.*

By Example 21 in Sadeghi and Soo [2022], it can be seen that combined, these conditions are strictly weaker than *restricted faithfulness*, which is the weakest known consistency condition for SGS/PC [Raskutti and Uhler, 2018].

Under the faithfulness assumption, constraint-based structure learning algorithms are natural. However, it is unclear whether these algorithms are still natural structure learning algorithms once the faithfulness assumption is relax, and no concrete algorithm is provided in Sadeghi and Soo [2022]. Thus, without assuming faithfulness, we aim to provide a general concrete natural structure learning algorithm that relaxes the consistency conditions in Proposition 1.

As usual in constraint-based causal learning, we assume the availability of a *conditional independence oracle*—given a probability distribution $P$, we can determine with certainty whether conditional independence statements are true. In practice, conditional independence statements need to be estimated from the data using methods such as HSIC testing [Gretton et al., 2007], and is shown to be in general, a hard problem [Shah and Peters, 2020].

## 3 THEORY AND METHODS

Here, we present our relaxation of the theory of natural structure learning algorithms and the practical algorithm.

### 3.1 THEORY

**Definition 8** (V-OUS and collider-stability). *A distribution $P$ is V-OUS and collider-stable wrt DAG $G$ if for all v-configuration $i \sim k \sim j$ in $G$:*

1. *(V-Ordered upward stability (V-OUS)). If $i \sim k \sim j$ is a non-collider, then for all $C \subseteq V \backslash \{i, j, k\}$, we have $i \perp\!\!\!\perp j \mid C \Rightarrow i \perp\!\!\!\perp j \mid C \cup \{k\}$.*

2. *(Collider-stability). If $i \to k \leftarrow j$, then there exists $C' \subseteq V \backslash \{i, j, k\}$ such that $i \perp\!\!\!\perp j \mid C'$.*

Collider-stability is related to *orientation faithfulness*, which states that the graph is faithful up to v-configurations in the graph [Ramsey et al., 2006]. However collider-stability is much weaker, even than the Markovian assumption.

**Proposition 2** (Collider-stable is very weak). *If $P$ is Markovian to $G$, then $P$ is collider-stable wrt $G$.*

V-OUS and collider-stability can be seen as local versions of ordered stabilities for the purposes of learning DAGs. In the case of DAGs, V-OUS can be seen as a relaxation of OUS since the implication $i \perp\!\!\!\perp j \mid C \Rightarrow i \perp\!\!\!\perp j \mid C \cup \{k\}$ in Definition 8 only has to hold for $i, j, k$ that are non-colliders in $G$. Likewise, collider-stable is implied by ODS and can be seen as a relaxation.

**Definition 9** (Localised Natural Structure learning (LoNS) algorithm). *An algorithm that takes input distribution $P$, and outputs $G(P)$ is localised natural if:*

1. *$\mathrm{sk}(P) = \mathrm{sk}(G(P))$.*

2. *$P$ is V-OUS and collider-stable wrt $G(P)$.*

Note that the above is the same with natural structure learning algorithms, just that one of the requirements is relaxed, namely Condition 2 in Definition 9. Thus, just like natural structure learning algorithms, all constraint-based algorithms that work under faithfulness are localised natural.

To characterise all DAGs that could be the output of a LoNS algorithm, we introduce the following orientation rule:

**Definition 10** (V-OUS and collider-stable orientation rule wrt $P$). *A V-OUS and collider-stable orientation rule wrt $P$ is defined as an assignment of v-configurations $i \sim k \sim j$ in $\mathrm{sk}(P)$ into colliders and non-colliders as follows:*

1. *If $i \perp\!\!\!\perp j \mid C$ and $i \not\perp\!\!\!\perp j \mid C \cup \{k\}$ for some $C \subseteq V \backslash \{i, j, k\}$, then assign $i \sim k \sim j$ to be a collider.*

2. *If for all $C$ such that $i \perp\!\!\!\perp j \mid C$, we have $k \in C$, then assign $i \sim k \sim j$ to be a non-collider.*

A DAG $G$ is said to satisfy the V-OUS and collider-stable orientation rule wrt $P$, if $G$ satisfies:

1. $\mathrm{sk}(P) = \mathrm{sk}(G)$.

2. For all v-configurations $i \sim k \sim j$ in $G$, via the orientation rule in Definition 10,

   > if $i \sim k \sim j$ is assigned to be a collider or non-collider, then $i \sim k \sim j$ is a collider or non-collider, respectively in $G$.

We have the following characterisation:

**Proposition 3** (Characterisation). *The DAG $G$ satisfies the V-OUS and collider-stable orientation rule wrt $P$, if and only if $P$ satisfies:*

*1.* $\mathrm{sk}(P) = \mathrm{sk}(G)$.

*2. P is V-OUS and collider-stable wrt G.*

We can now apply the V-OUS and collider-stable orientation rule wrt $P$ to assign the v-configurations in $\mathrm{sk}(P)$. Note that the assignment may be incomplete, in the sense that some v-configurations may not satisfy either of the conditions in Definition 10 and are therefore unassigned. *Modified V-stability* is then defined as when this ambiguity can be resolved using the constraint that the graph is a DAG, as illustrated in Figure 1.

**Definition 11** (Modified V-stability). *A distribution $P$ is* modified V-stable*, if the v-configurations of DAGs that satisfy the V-OUS and collider-stable orientation rule wrt $P$ is unique.*

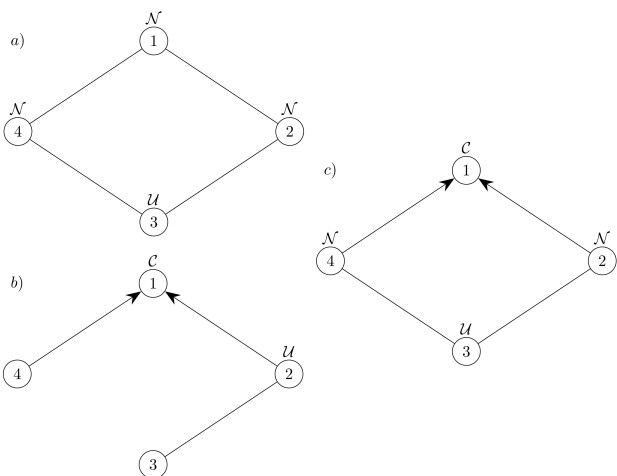

Figure 1: Different assigned $\mathrm{sk}(P)$. Label $\mathcal{N}, \mathcal{C}, \mathcal{U}$ denotes the v-configuration that is assigned to be a non-collider, collider and unassigned, respectively. In a), the unassigned v-configuration $4 \sim 3 \sim 2$ is constrained to be a collider due to acylicity of a DAG. In b), from lack of bidirectedness in a DAG, unassigned v-configuration $1 \sim 2 \sim 3$ is constrained to be a non-collider. In c), since there exists DAGs such that v-configuration $2 \sim 3 \sim 4$ can be either a collider or a non-collider, this ambiguity cannot be resolved.
$P$ is modified V-stable if, after orienting $\mathrm{sk}(P)$ via the V-OUS and collider-stable orientation rule wrt $P$, the unassigned v-configurations in $\mathrm{sk}(P)$ can be resolved using DAG constraints, as in a) and b).

Under V-stability, all v-configurations in $\mathrm{sk}(P)$ must satisfy either of the conditions in Definition 10, leaving no v-configurations in $\mathrm{sk}(P)$ unassigned. Thus, V-stability implies modified V-stability.

**Remark 4.** *Combined with Proposition 3, we have the following equivalent notion of modified V-stability: all DAGs $G$ to which $P$ satisfies:*

*1.* $\mathrm{sk}(P) = \mathrm{sk}(G)$.

*2. P is V-OUS and collider-stable wrt G.*

*are Markov equivalent.*

We have the following, for the true causal graph $G_0$:

**Theorem 1** (Sufficient and necessary consistency conditions for LoNS). *LoNS algorithms are consistent if and only if:*

*1. $P$ satisfies adjacency faithfulness wrt $G_0$.*

*2. $P$ is V-OUS and collider-stable wrt $G_0$.*

*3. $P$ is modified V-stable.*

The following example shows that even when combined with $P$ being adjacency faithful and V-OUS and collider-stable wrt $G_0$, V-stability of $P$ need not be implied:

**Example 1.** *Let $G_0$ be $1 \to 2 \to 3 \leftarrow 4$, and $P$ induces all the conditional independence implied from the Markov property wrt $G_0$ in addition to $1 \perp\!\!\!\perp 3$.*

*The v-configuration $2 \sim 3 \sim 4$ in $\mathrm{sk}(P)$ satisfies $2 \perp\!\!\!\perp 4$ and $2 \not\perp\!\!\!\perp 4 \mid 3$, then after $2 \sim 3 \sim 4$ is assigned as a collider, this constraints $1 \sim 2 \sim 3$ to be a non-collider. Thus $P$ is modified V-stable. Adjacency faithfulness is obvious.*

*Since we have $1 \perp\!\!\!\perp 3$ and $1 \perp\!\!\!\perp 3 \mid 2$, we have that $P$ is not V-stable, but V-OUS holds since $1 \perp\!\!\!\perp 3 \mid \{2, 4\}$.*

Thus the conditions in Theorem 1 is strictly weaker than those in Proposition 1, which is already weaker than restricted faithfulness, and we will see in Section 4 that these conditions are different to the sufficient and necessary conditions of SP.

### 3.1.1 Realising and Interpreting the V-OUS Condition

V-OUS is implied by faithfulness. Here, without assuming faithfulness, we discuss cases in which the V-OUS property can still arise, and provide basic interpretations.

**Proposition 4** (Conditional exchangability and composition imply V-OUS). *Let $P$ satisfy:*

*1. (Composition property). For all disjoint $i, j, k, C$, the following holds: $i \perp\!\!\!\perp j \mid C$ & $i \perp\!\!\!\perp k \mid C \Rightarrow i \perp\!\!\!\perp \{j, k\} \mid C$.*

*2. (Conditional exchangability). For all non-collider v-configuration $i \sim k \sim j$ in $G_0$, the marginal distribution of $P$ on $\{i, j, k\}$ conditioned on $V \backslash \{i, j, k\}$ is exchangable.*

*Then $P$ satisfies V-OUS wrt $G_0$.*

The composition property allows the deduction of joint independence from pairwise independence, and is satisfied

by some common distributions such as Gaussians. The exchangability assumption is commonly made when nodes are indistinguishable from one another, such as in Bayesian theory.

The V-OUS assumption can be interpreted as a prevention of Simpson's paradox on non-collider v-configuration $i \sim k \sim j$, since all conditional independencies of $i$ and $j$ are preserved when conditioning on $k$.

## 3.2 CONSTRUCTION OF A LONS ALGORITHM

Having described the LoNS algorithms, we provide a pseudocode of such an algorithm:

---
**Algorithm 1** **M**odified **V**-stable **Lo**calised **N**atural **S**tructure Learning (Me-LoNS)

---
**Input**: Probability distribution $P$
**Output**: DAG $G(P)$

1: Construct $\text{sk}(P)$.
2: Apply the V-OUS and collider-stable orientation rule wrt $P$ to assign v-configurations in $\text{sk}(P)$.
3: Solve for a DAG $G(P)$ having skeleton $\text{sk}(P)$ and satisfy the assignment of v-configurations. If no solution exists, **return** error.
4: **return** $G(P)$.

---

**Remark 5.** *Generally, Me-LoNS differs from PC [Spirtes et al., 2000] only in determining whether v-configurations in $\text{sk}(P)$ is a collider.*

*Since we have to check conditional independence statements of all subsets, the algorithm have exponential time complexity which is comparable to the skeleton building step of PC, and is a big improvement compared to the factorial running time of SP.*

*Since the running time of greedy versions of SP based on depth-first search [Solus et al., 2021, Lam et al., 2022] are generally dependent on the depth parameter, it is not obvious whether the running time of Me-LoNS is an improvement.*

Note that Me-LoNS outputs a DAG, and since in the observational causal learning setting we are interested in the corresponding MEC, we can always convert the DAG into CP-DAG which is a graphical object uniquely representing a MEC, for example via the `dag2cpdag` function in the `causal-learn` Python package [Zheng et al., 2024].

**Proposition 5** (Me-LoNS is a LoNS algorithm). *Me-LoNS is a LoNS algorithm if and only if there exists a DAG $G$ to which $P$ satisfies the following:*

1. $\text{sk}(P) = \text{sk}(G)$.
2. *$P$ is V-OUS and collider-stable wrt $G$.*

The consistency conditions of Me-LoNS is then given in Theorem 1. Note that modified V-stability of input distribution $P$ ensures that the output of Me-LoNS is unique up to MEC.

**Remark 6.** *The orientation rule of Me-LoNS is similar to conservative PC (CPC) [Ramsey et al., 2006] in the sense that:*

1. *Both assign non-colliders similarly.*
2. *Both allow for ambiguous or unassigned v-configurations.*
3. *Both have a criterion when the consistency condition relating distribution $P$ and true causal graph $G_0$ is violated; if Me-LoNS errors, there is no DAG that satisfies the conditions in Proposition 5 (by applying the characterisation in Proposition 3).*

*However, Me-LoNS relaxes the restricted-faithfulness condition of CPC by orienting colliders differently.*

# 4 SIMULATION AND THEORETICAL COMPARISONS

We will compare the consistency conditions of Me-LoNS to some existing constraint-based causal learning algorithms both theoretically and via simulations using the `causal-learn` package [Zheng et al., 2024] in Python. Me-LoNS is implemented via the following steps:

1. Use the same skeleton discovery function as PC.
2. Make a new orientation function based on the new orientation rule.
3. Use the `scipy.optimize` package to solve the DAG search problem.

We will be using mixed linear integer programming with a constant objective to solve Step 3 of Me-LoNS. In addition to the layered network (LN) formulation from Manzour et al. [2021], we introduce additional constraints from Step 2 of Me-LoNS as follows:

$$
\begin{aligned}
z_{ik} = z_{jk} = 1 & \qquad \forall i \sim k \sim j \in \mathcal{C} \\
z_{ik} + z_{jk} \leq 1 & \qquad \forall i \sim k \sim j \in \mathcal{N}
\end{aligned}
$$

where $z_{ij} = 1$ if $i \to j$, and $z_{ij} = 0$ if $i \leftarrow j$, and $\mathcal{C}, \mathcal{N}$ the set of v-configurations in $\text{sk}(P)$ that are assigned to be colliders and non-colliders respectively by the V-OUS and collider-stable orientation rule wrt $P$ in Step 2 of Me-LoNS.

Within each comparison, we will simulate data from the same structural equation model (SEM), with corresponding causal graph $G_0$ to obtain a total of 1,000,000 samples. These samples are then subdivided into 100 test units of 10,000 samples each. From these 100 tests, we compare

the percentage of tests the algorithms return the consistent output (output is Markov equivalent to $G_0$). Whenever conditional independence testing is needed, the `fisherz` conditional independence test from the package is used throughout with a significance of 0.05. To test for Markov equivalence of the true causal graph $G_0$ and the output graph of the algorithm, the `mec_check` function is used.

**Remark 7.** *Although Me-LoNS is deterministic, due to conditional independence testing being used in simulations, the simulation output is non-deterministic. The simulations aim to investigate how well the theoretical results (stated purely in conditional independencies) hold up under the randomness of conditional independence testing.*

## 4.1 COMPARISON TO THE PC ALGORITHM

Me-LoNS strictly generalises PC, as in the following:

**Proposition 6** (Me-LoNS strictly generalises PC). *If $P$ is V-stable, then the outputs of both PC and Me-LoNS are Markov equivalent. Furthermore, there exist distribution $P$ and true causal graph $G_0$, such that Me-LoNS is consistent but not PC.*

**Remark 8.** *In general, PC outputs a representative of a MEC (CP-DAG). Proposition 6 states that under V-stability, Me-LoNS returns a DAG that is of the MEC represented by the CP-DAG output of PC regardless of violations of consistency conditions.*

$$\epsilon_i \overset{i.i.d.}{\sim} N(0,1), i = 1, 2, 3, 4$$
$$X_1 = \epsilon_1$$
$$X_2 = \epsilon_2$$
$$X_3 = -6X_1 + 2X_2 + \epsilon_3$$
$$X_4 = 3X_1 + 4X_2 + \epsilon_4 \qquad (1)$$

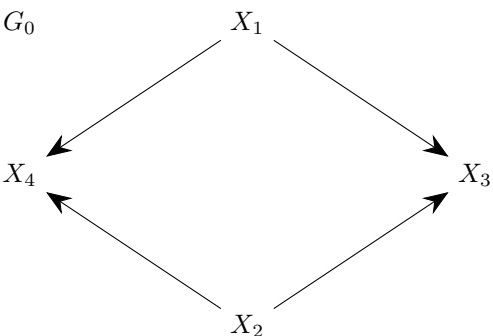

Figure 2: SEM 1 corresponds to the DAG $G_0$.

Table 1: Percentage of simulations from SEM 1 that the algorithm returns a consistent output.

| PC | Me-LoNS |
|----|---------|
| 8% | 90% |

To illustrate Proposition 6, we compare Me-LoNS with the PC algorithm from the package using the `definiteMaxP` orientation rule which orients only definite colliders and definite non-colliders (thus in this setting, PC coincides with CPC). The input distribution $P$ will be induced by SEM 1, having all the conditional independencies implied by the Markovian property wrt $G_0$ in Figure 2, in addition to $X_1 \perp\!\!\!\perp X_2 \,|\, \{X_3, X_4\}$.

PC fails to identify the colliders in Figure 2, due to violation of orientation faithfulness. This is reflected in Table 1.

## 4.2 COMPARISON TO SPARSEST PERMUTATION (SP) ALGORITHM

The *Sparsest Markov Representation (SMR)* assumption is the sufficient and necessary consistency condition for SP [Raskutti and Uhler, 2018], and it is strictly different to the consistency conditions of Me-LoNS in Theorem 1, as the following example shows:

**Example 2** (Me-LoNS and SP are different/incomparable). *(SMR, but not conditions in Theorem 1). Consider the example from Raskutti and Uhler [2018]. Let $G_0$ be the following:*

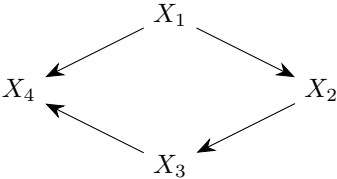

*and $P$ implies the conditional independencies $X_1 \perp\!\!\!\perp X_3 \,|\, X_2$ and $X_2 \perp\!\!\!\perp X_4 \,|\, \{X_1, X_3\}$ and $X_1 \perp\!\!\!\perp X_2 \,|\, X_4$. It can be seen that SMR holds, but adjacency faithfulness is violated.*

*(Conditions in Theorem 1, but not SMR). Consider the graph $G_0$ in Figure 3, with $P$ implying the conditional independencies $X_2 \perp\!\!\!\perp X_3$ and $X_1 \perp\!\!\!\perp X_3$ and $X_2 \perp\!\!\!\perp X_3 \,|\, \{X_1, X_4\}$. Here, adjacency faithfulness holds. V-OUS holds since there are no non-colliders to check in $G_0$, and modified V-stability of $P$ also holds since V-stability of $P$ holds.*

*$P$ is Markovian to both $G_0$ and $G'$ where $G'$ differs from $G_0$ by flipping the edge $X_2 \to X_4$. $G_0$ and $G'$ are both sparsest Markovian graphs to $P$, but are not Markov equivalent, thus SMR does not hold.*

*Note that this counter-example hinges on the fact that*

*singleton transitivity of $P$ does not hold otherwise we would have $X_2 \perp\!\!\!\perp X_4 \mid X_1$ or $X_3 \perp\!\!\!\perp X_4 \perp\!\!\!\perp X_1$, violating adjacency faithfulness, thus $P$ cannot be Gaussian.*

$$\epsilon_i, \phi_j \overset{\text{i.i.d.}}{\sim} \text{Bern}(\tfrac{1}{2}), i = 1, \ldots, 4, j = 1, \ldots, 5$$
$$X_1 = (\phi_1, \phi_2, \epsilon_1)$$
$$X_2 = (X_1^1, \phi_3, \epsilon_2)$$
$$X_3 = (\phi_4, \phi_5, \epsilon_3)$$
$$X_4 = (X_1^1 + X_3^1, X_2^1 + X_3^2, X_2^2, \epsilon_4) \qquad (2)$$

Here the $+$ in the structural assignment of $X_4$ in SEM 2 denotes regular addition, and $X_i^j$ denotes the $j$-th entry from the left of $X_i$.

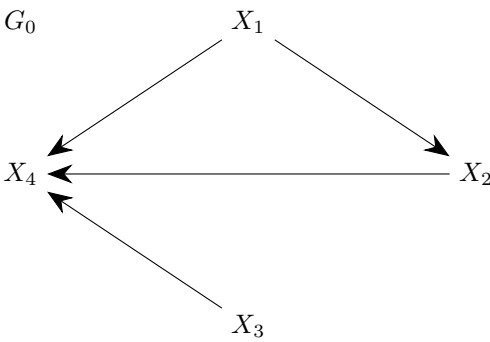

Figure 3: SEM 2 corresponds to the DAG $G_0$.

**Remark 9.** *In the supplementary material, $X_1 \perp\!\!\!\perp X_3 \mid X_2$ and $X_2 \perp\!\!\!\perp X_3 \mid X_1$ are not needed for the example, and is merely a byproduct from the construction of SEM 2.*

Since greedy versions of the SP algorithm have stronger consistency conditions, Example 2 shows that Me-LoNS is a viable alternative to all greedy variants of SP since Me-LoNS works under different conditions.

To illustrate the incomparability of Me-LoNS and SP from Example 2, we compare Me-LoNS to the implementation of SP in the package, *greedy relaxation of sparsest permutation (GRaSP)* Lam et al. [2022]. The input distribution $P$ will be induced by SEM 2, having all the conditional independencies in Example 2 in addition to $X_2 \perp\!\!\!\perp X_3 \mid X_1$ and $X_1 \perp\!\!\!\perp X_3 \mid X_2$.

**Remark 10.** *GRaSP cannot differentiate the direction of edge $X_2 \to X_4$ in Figure 3, thus it returns a consistent output about half the time, as shown in Table 2. In the case of $G_0$ being comprised of $n$ disconnected components, with each component being the $G_0$ in Figure 3, GRaSP will then return a consistent output about $\frac{1}{2^n}$ of the time.*

Table 2: Percentage of simulations from SEM 2 that the algorithm returns a consistent output.

| GRaSP | Me-LoNS |
|-------|---------|
| 56%   | 94%     |

## 5  CONCLUSION AND FUTURE WORK

The contributions of this paper can be summarised in Figure 4:

The proposed Me-LoNS algorithm has the following desirable properties:

1. It is a strict generalisation the PC algorithm, and is consistent under strictly different conditions than SP.

2. It has exponential run time which is comparable to the skeleton building step of SGS.

Hence, Me-LoNS provides another option for an algorithm that is consistent strictly beyond faithfulness, but runs in exponential time which is better than the factorial running time of SP algorithm [Raskutti and Uhler, 2018]. Although there exist speed-ups of the SP algorithm, such as ones based on greedy search like GRaSP used in the Section 4, these algorithms are faster at the cost of stronger consistency conditions [Solus et al., 2021, Lam et al., 2022].

Note that the work done is focused on DAGs, it may be possible to extend the work done to ancestral graphs, which represents causal systems with latent variables, since the notion of ordered upward and downward stabilities are well defined for anterial graphs in general [Sadeghi, 2017].

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

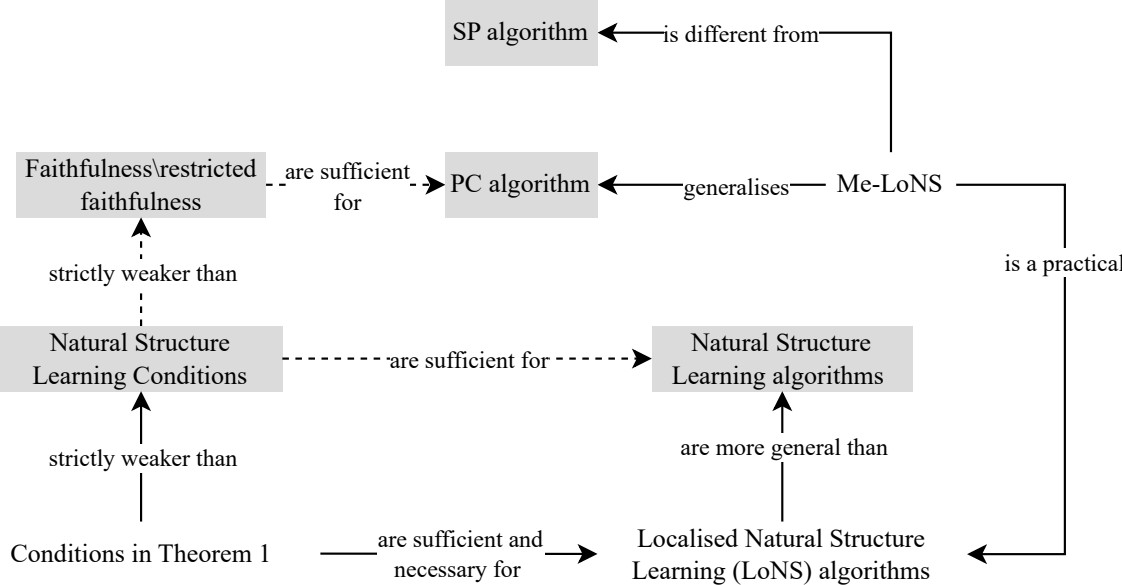

Figure 4: Diagram relating the results of this paper, with the dotted arrows and shaded texts indicating existing results and the rest of diagram being novel contributions.

Joseph D. Ramsey, Jiji Zhang, and Peter Spirtes. Adjacency-faithfulness and conservative causal inference. In *UAI '06, Proceedings of the 22nd Conference in Uncertainty in Artificial Intelligence, Cambridge, MA, USA, July 13-16, 2006*. AUAI Press, 2006.

Garvesh Raskutti and Caroline Uhler. Learning directed acyclic graph models based on sparsest permutations. *Stat*, 7(1):e183, 2018. e183 sta4.183.

Kayvan Sadeghi. Faithfulness of probability distributions and graphs. *J. Mach. Learn. Res.*, 18(148):1–29, 2017.

Kayvan Sadeghi and Terry Soo. Conditions and assumptions for constraint-based causal structure learning. *J. Mach. Learn. Res.*, 23(109):1–34, 2022.

Rajen D. Shah and Jonas Peters. The hardness of conditional independence testing and the generalised covariance measure. *Ann. Statist.*, 48(3):1514 – 1538, 2020.

L Solus, Y Wang, and C Uhler. Consistency guarantees for greedy permutation-based causal inference algorithms. *Biometrika*, 108(4):795–814, 01 2021.

Peter Spirtes, Clark Glymour, and Richard Scheines. MIT Press, 2000.

Caroline Uhler, Garvesh Raskutti, Peter Bühlmann, and Bin Yu. Geometry of faithfulness assumption in causal inference. *Ann. Statist.*, 41:436–463, 2013.

Jiji Zhang and Peter Spirtes. Detection of unfaithfulness and robust causal inference. *Minds and Machines*, pages 239–271, 2008.

Yujia Zheng, Biwei Huang, Wei Chen, Joseph Ramsey, Mingming Gong, Ruichu Cai, Shohei Shimizu, Peter Spirtes, and Kun Zhang. Causal-learn: Causal discovery in Python. *J. Mach. Learn. Res.*, 25(60):1–8, 2024.

# Localised Natural Causal Learning Algorithms for Weak Consistency Conditions (Supplementary Material)

**Kai Z. Teh**[1]  **Kayvan Sadeghi**[1]  **Terry Soo**[1]

[1]Department of Statistical Science, UCL, London, UK

## A  PROOFS

We will use the following well known results:

**Proposition 7** ([Lauritzen, 1996]). *If $P$ is Markovian to $G$, then $P$ is pairwise Markovian to $G$; that is, for every non-adjacent $i, j$, we have $i \perp\!\!\!\perp j \mid \mathrm{an}_G(i, j)$.*

**Proposition 8** ([Verma and Pearl, 1990]). *The DAGs $G_1$ and $G_2$ are Markov equivalent if and only if:*

1. $\mathrm{sk}(G_1) = \mathrm{sk}(G_2)$.
2. *The set of colliders in $G_1$ coincides with the set of colliders in $G_2$.*

**Proposition 9** ([Ramsey et al., 2006]). $\mathrm{sk}(P) = \mathrm{sk}(G)$ *if and only if $P$ is adjacency faithful wrt $G$.*

*Proof of Proposition 2.* Let $i, k, j$ be a collider in the DAG $G$. Since $P$ is Markovian to $G$, we have that $i \perp\!\!\!\perp j \mid \mathrm{an}_G(i, j)$ by the pairwise Markov property in Proposition 7, and by acyclicity of $G$, we have $k \notin \mathrm{an}_G(i, j)$.  □

*Proof of Proposition 3.*
**If**: Let $P$ be V-OUS and collider-stable wrt $G$. Since $\mathrm{sk}(P) = \mathrm{sk}(G)$, it suffices to show Item 2 that, for v-configurations $i \sim k \sim j$ in $\mathrm{sk}(P) = \mathrm{sk}(G)$, we have:

1. If $i \sim k \sim j$ is assigned to be a collider, then if $i \sim k \sim j$ is a non-collider in $G$, V-OUS is violated.
2. Likewise if $i \sim k \sim j$ is assigned to be a non-collider $k \in C$, then if $i \sim k \sim j$ is a collider in $G$, collider-stability is violated.

Note that this is due to the orientation rules in Definition 10 being negations of the V-OUS and collider-stability property. Thus $G$ satisfies the orientation rule wrt $P$.

**Only if**: Let G satisfy the V-OUS and collider-stable orientation rules wrt $P$. Since then for v-configurations $i \sim k \sim j$ in $G$: From Item 2, we have the following breakdown:

1. Let $i \sim k \sim j$ is a collider in $G$. It is easy to verify that in both cases, where it is assigned as a collider or remains unassigned, that $i \sim k \sim j$ is collider-stable wrt $G$.
2. Let $i \sim k \sim j$ be a non-collider in $G$.
   (a) If $i \sim k \sim j$ is assigned as a non-collider, then $i \sim k \sim j$ is V-OUS wrt $G$.
   (b) Let $i \sim k \sim j$ be unassigned, and $C \subseteq V \setminus \{i, j, k\}$. If $i \perp\!\!\!\perp j \mid C$, then we must also have $i \perp\!\!\!\perp j \mid C \cup \{k\}$, otherwise $i \sim k \sim j$ would have been assigned to be a collider. Hence $i \sim k \sim j$ is V-OUS wrt $G$.

Thus $P$ is V-OUS and collider-stable wrt $G$, and $\mathrm{sk}(P) = \mathrm{sk}(G)$ follows from Item 1.  □

To show Theorem 1, we first show Remark 4:

*Proof of Remark 4.* By Proposition 3 we see that $P$ being modified V-stable is equivalent to: DAGs $G$ to which $P$ satisfies $\mathrm{sk}(P) = \mathrm{sk}(G)$, and V-OUS and collider-stable wrt have unique v-configurations. This is equivalent to all such DAGs are Markov equivalent by Proposition 8. $\square$

**Proposition 10.** *Let $G_1$ and $G_2$ be Markov equivalent DAGs, with the same distribution $P$. If $P$ is V-OUS and collider-stable wrt $G_1$, then $P$ is V-OUS and collider-stable wrt $G_2$.*

*Proof.* By Proposition 8, $\mathrm{sk}(G_1) = \mathrm{sk}(G_2)$ and the v-configurations and colliders in $G_1$ and $G_2$ coincide. Since $P$ is the same for both $G_1$ and $G_2$, $P$ is V-OUS and collider-stable wrt $G_2$ by virtue that it is for $G_1$. $\square$

*Proof of Theorem 1.* Denote the output of the algorithm by $G(P)$.
**If**: Since $P$ is adjacency faithful wrt $G_0$, by Proposition 9 and definition of LoNS, $\mathrm{sk}(P) = \mathrm{sk}(G) = \mathrm{sk}(G(P))$. $P$ is also V-OUS and collider-stable wrt both $G_0$ and $G(P)$, thus $G_0$ is Markov equivalent to $G(P)$ by Remark 4.
**Only if**: Let $G(P)$ and $G_0$ be Markov equivalent.

1. (Adjacency faithfulness). By Proposition 8, and since $G(P)$ is Markov equivalent to $G_0$, we have that $\mathrm{sk}(P) = \mathrm{sk}(G(P)) = \mathrm{sk}(G_0)$, and by Proposition 9, adjacency faithfulness follows.
2. (V-OUS and collider-stable). Since $P$ is V-OUS and collider-stable wrt $G(P)$, by Proposition 10, $P$ is V-OUS and collider-stable wrt $G_0$.
3. (Modified V-stability). Appealing to Remark 4, we note that any DAG satisfying the conditions in the remark is the output of a LoNS algorithm; thus by assumption, these DAGS are Markov equivalent to $G_0$. $\square$

To show Proposition 4, we use the following from Sadeghi [2020]:

**Proposition 11** ([Sadeghi, 2020]). *If $P$ is exchangable, then $P$ satisfying composition is equivalent to $P$ satisfying upward stability; that is, for all $i, j, k \in V$, we have $i \perp\!\!\!\perp j \mid C \Rightarrow i \perp\!\!\!\perp j \mid C \cup \{k\}$.*

*Proof of Proposition 4.* For non-collider $i \sim k \sim j$ in $G$, exchangability of the marginal $\{i, j, k\}$ given $C \subseteq V \backslash \{i, j, k\}$ follows from conditional exchangability. Combined with Proposition 11 and composition, the marginal of $\{i, j, k\}$ conditional on any $C \subseteq V \backslash \{i, j, k\}$ is upward-stable, thus implying V-OUS. $\square$

*Proof of Proposition 5.*
**If:** Since there exists a DAG $G$ to which $P$ satisfies V-OUS and collider-stability and $\mathrm{sk}(P) = \mathrm{sk}(G)$, Proposition 3 guarantees that a DAG that satisfies the V-OUS and collider-stable orientation rule wrt $P$ exists, and will be returned by Step-3 of Me-LoNS, again by Proposition 3, $P$ will be adajcency faithful and V-OUS and collider-stable wrt this output.

**Only if:** If there does not exist a DAG $G$ to which $P$ satisfies V-OUS and collider-stability and $\mathrm{sk}(P) = \mathrm{sk}(G)$, by Proposition 3, there is no DAG that satisfies the V-OUS and collider-stable orientation rule wrt $P$, thus Step 3 of Me-LoNS errors. $\square$

*Proof of Proposition 6.*

1. (Me-LoNS generalises PC under V-stability). Under V-stability of $P$, the V-OUS and collider-stable orientation rule wrt $P$ for assigning colliders becomes the following: for $i \sim k \sim j$ in $\mathrm{sk}(P)$, we have

$$\exists C \subseteq V \backslash \{i, j, k\} \quad i \perp\!\!\!\perp j \mid C \ \& \ i \not\perp\!\!\!\perp j \mid C \cup \{k\} \iff \exists C \subseteq V \backslash \{i, j, k\} \quad i \perp\!\!\!\perp j \mid C.$$

Note that the RHS is the negation of the V-OUS and collider-stable orientation rule wrt $P$ when assigning a non-collider, thus the orientation rules reduce to the following:

   (a) If $k \in C$ for all $C$ such that $i \perp\!\!\!\perp j \mid C$, then assign $i \sim k \sim j$ to be a non-collider (unchanged).
   (b) Otherwise, assign $i \sim k \sim j$ to be a collider.

This is the same as in PC.

2. (Me-LoNS works but not PC). Consider SEM 1, which gives the graph $G_0$ in Figure 2 with the set of conditional independencies being $X_1 \perp\!\!\!\perp X_2$ and $X_1 \perp\!\!\!\perp X_2 \,|\, \{X_3, X_4\}$ and $X_3 \perp\!\!\!\perp X_4 \,|\, \{X_1, X_2\}$. Thus we see that the V-OUS-collider-stable orientation rule orients all the v-configurations correctly, but PC orients the collider $X_1 \to X_3 \leftarrow X_2$ as a non-collider. $\qquad\square$

**References (Supplementary Material)**

Steffen L. Lauritzen. *Graphical Models*. Oxford University Press, 1996.

Joseph D. Ramsey, Jiji Zhang, and Peter Spirtes. Adjacency-faithfulness and conservative causal inference. In *UAI '06, Proceedings of the 22nd Conference in Uncertainty in Artificial Intelligence, Cambridge, MA, USA, July 13-16, 2006*. AUAI Press, 2006.

Kayvan Sadeghi. On finite exchangeability and conditional independence. *Electron. J. Stat.*, 14(2):2773 – 2797, 2020.

Tom S. Verma and Judea Pearl. On the equivalence of causal models. In *Proceedings of the Sixth Conference on Uncertainty in Artificial Intelligence*, pages 220–227. Elsevier Science, 1990.
