# OpenReview forum: "Localised Natural Causal Learning Algorithms for Weak Consistency Conditions"
_auai.org/UAI/2024/Conference — UAI 2024 poster_

### Official Review · Reviewer_n2D5 · 2024-03-11

**Q2-1 Originality-Novelty:** 2
**Q2-2 Correctness-Technical Quality:** 3
**Q2-5 Clarity Of Writing:** 3

**Q10 Ethical Concerns:**

There are no ethical concerns

**Q1 Summary And Contributions:**

**Short Summary:**

This paper defines the notion of a localised natural structure learning (LoNS) algorithm, which relaxes the previously defined notion of a natural structure learning algorithm. It is shown that LoNS algorithms recover the Markov equivalence class of the true DAG from tests of conditional independence by using a condition that is strictly weaker than the faithfulness condition.

**Long Summary/Contributions:**

This paper looks at the setting of constraint-based learning of the causal structure. If the Markov property is assumed to hold, then popular algorithms such as PC additionally require the faithfulness assumption to consistently find the Markov equivalence class (MEC) of the true DAG.

This paper revisits the notion of natural structure learning algorithms, which are algorithms that output a graph $G$ given a probability distribution $P$ such that that the $G$ satisfies certain properties w.r.t. $P$ (such as skeleton soundness and ordered upward/downward stability). Natural structure learning algorithms have been proven to soundly find the MEC of the true DAG.
Constraint-based algorithms like PC are natural structure learning algorithms, even though the consistency of natural structure learning hinges on assumptions that are strictly weaker than faithfulness (previous work).

In introducing LoNS algorithms, that are shown to be a superset of natural structure learning algorithms, the paper is able to establish even weaker conditions for consistency.

**Q2-3 Extent To Which Claims Are Supported By Evidence:**

2: Fair: the main claims are somewhat supported by evidence (but the experimental evaluation may be weak, or does not match entirely with the claims, important baselines may be missing, proofs contain important ideas but lack rigor, algorithmic details are only discussed superficially, references are imprecise, assumptions are not sufficiently motivated or explicated, etc.).

**Q2-4 Reproducibility:**

3: Good: key resources (e.g. proofs, code, data) are available and key details (e.g. proofs, experimental setup) are sufficiently well-described for competent researchers to confidently reproduce the main results.

**Q3 Main Strengths:**

1. This is a concise and easy-to-read paper, that supports all statements with proofs.

2. It provides the background knowledge with the right amount of detail to follow the novel ideas on a new class of algorithms and why these algorithms require weaker conditions for consistency.

3. In particular, this work weakens the faithfulness assumption, required for constraint-based causal structure learning, even further compared to previous works, and shows how a new class of algorithms can consistently discover the MEC when this weaker faithfulness holds.

**Q4 Main Weakness:**

1. The central weakness of this paper is the simulation section which compares the performance of Me-LoNS algorithm with PC and GRaSP algorithms for a select structural equation model each. That is to say, the two examples are picked by hand, and while these illustrative examples help to understand the theoretical differences between these algorithms, they cannot be considered representative of an overall markedly better performance. This can only be seen in simulations with a sufficient number of random DAGs.

2. While it is of theoretical interest that faithfulness can be relaxed and the MEC is still discoverable, are violations of the new weaker conditions (V-OUS and collider stability) detectable? One strength of the conservative-PC (CPC) algorithm, even though its output contains ambiguous e-patterns, is that it can mark where orientation faithfulness is violated. The question: are violations of this weaker orientation faithfulness detectable, and is the detection any different from the one employed in CPC? Because if not, I constitute it to be a weakness in applicability of the LoNS idea.

3. There are certain formulation errors, that I point out in detailed comments to the author.

**Q5 Detailed Comments To The Authors:**

**General Comments:**

1. In the Simulation section, what is the interpretation for the PC algorithm performing so much worse? If the incorrect conditional independence (CI) test $X_1 \perp X_2 |\  (X_3, X_4)$ only happens the cardinality of conditioning set equalling 2, and the edge between $X_1$ and $X_2$ has been unconditionally removed, why would the incorrect CI test influence the result of PC? Unless, you are re-testing for conditioning sets as conservative PC does, in which case, this should be underlined.

2. How is performance defined in the simulation section in tables 1 and 2?

**Specific Comments:**

1. In Section 1, first paragraph, restricted-SCM and invariant causal prediction based approaches for structure learning also merit mention.
2. In Section 1, first paragraph, first sentence is unclear. Assumptions are necessary for constraint-based structure learning, so the clause after "otherwise" is not entirely correct.
3. Proposition 3 is not clear, it seems to me to be a recursive statement.
4. The conservative PC (CPC) algorithm merits a mention in the main body of this work, not just the appendix, as that work lays down the notions of adjacency and orientation faithfulness and uses them to define CPC which is a consistent algorithm. It admittedly does not recover the MEC due to the admission of e-patterns that violate orientation faithfulness, but is shown to output much fewer wrong causal arrowheads as opposed to PC.

**Typos/Grammar/Notational Issues:**

1. In Section 3.1.1, the first sentence is grammatically incorrect, and possibly incomplete.
2. In Section 3.1.1, in the second sentence, the 'at' should be replaced by 'in'.
3. In 3.1.1., the last sentence of the second last paragraph is incomplete.
4. In Section 5, the last sentence, 'ancestral' instead of 'anterial'.
5.  The sentence above Proposition 2 is grammatically incorrect and sounds vague.

**Q9 Complying With Reviewing Instructions:**

Yes

---

> ### Author Rebuttal · Authors · 2024-04-04
>
> Thank you for the insightful feedback, please find the following addressing your comments:
>
> Q4
>
> 1. The simulation section highlights the cases where Me-LoNS recovers the correct graph over other algorithms. See Example 2 on page 6 where we ``consider the example in the proof of Theorem 2.4 in Raskutti and Uhler [2018] in which SMR holds but adjacency faithfulness is violated.’’ for the case when Me-LoNS fails. While running the simulations, we have also conducted some tests where the output of Me-LoNS coincides with PC for some benchmark datasets. This could be included in the final revision as we have plenty of space left.
>
> 2. Yes, the violation is detectable via step 3 of the algorithm on page 5 when an error is returned when no DAG solution exists.
>
> Q5
>
> General
>
> 1. Due to not identifying the colliders in the graph, PC performs worse, as summarised on page 6 ''PC fails to identify the colliders in Figure 2''. We also mention ``from the package using the definiteMaxP orientation rule'' on page 6, explaining the setting of our simulation, from which the definiteMaxP orientation rule from Python package causal-learn is similar to Conservative PC. We will make this link clearer in the final revision since this may be unclear to those unfamiliar with running the package.
>
> 2. The performance is defined by the percentage of tests (out of 100) that returns the correct/consistent output by each algorithm.
>
> Specific
>
> 3. Proposition 3 relates graphs satisfying the orientation rule with graphs satisfying the given conditions. Our orientation rule will be re-named in the final revision to avoid this confusion.
>
> 4. Yes, we will make more short but explicit connections to the CPC algorithm in the final revision.
>
> Typos/Grammar
>
> Thank you for alerting us, we will correct these in the final revision.

---

### Official Review · Reviewer_n4m2 · 2024-03-17

**Q2-1 Originality-Novelty:** 2
**Q2-2 Correctness-Technical Quality:** 2
**Q2-5 Clarity Of Writing:** 3

**Q10 Ethical Concerns:**

No.

**Q1 Summary And Contributions:**

Following Sadeghi (2017) in characterizing a natural structure learning algorithm, this work identifies a weaker set of consistency assumptions for causal discovery. Their generalization is highly intuitive in the sense of handling unshielded triples. This work brings an interesting discussion on how to recover the true causal mechanism under consistency assumptions strictly weaker than (restricted-)faithfulness that has been widely adopted by researchers.

**Q2-3 Extent To Which Claims Are Supported By Evidence:**

2: Fair: the main claims are somewhat supported by evidence (but the experimental evaluation may be weak, or does not match entirely with the claims, important baselines may be missing, proofs contain important ideas but lack rigor, algorithmic details are only discussed superficially, references are imprecise, assumptions are not sufficiently motivated or explicated, etc.).

**Q2-4 Reproducibility:**

2: Fair: key resources (e.g. proofs, code, data) are unavailable but key details (e.g. proof sketches, experimental setup) are sufficiently well-described for an expert to confidently reproduce the main results.

**Q3 Main Strengths:**

The main strength of this work lies in the generalization of Sadeghi (2017). While natural structure learning algorithms are a nice generalization of the standard constraint-based algorithms (e.g., PC), the authors argued for a strictly weaker set of consistency assumptions required to identify the true Markov Equivalence Class (MEC). Their generalization is highly intuitive and natural by targeting how unshielded triples (aka v-configuration) can be oriented by a strictly weaker set of rules, which is associated with a set of weaker consistency assumptions. Most proofs are written cleanly and concisely.

**Q4 Main Weakness:**

1. This work can easily be interpreted as a generalization of Ramsey et al. (2006) “Adjacency-Faithfulness and Conservative Causal Inference”. In short, conservative PC is motivated by the fact that unshielded triples cannot always be oriented unambiguously by PC. The former introduced conservative orientation rules so unshielded triples can be marked as ambiguous to represent inferential uncertainty. This work can be seen as a way to strengthen conservative PC by claiming that some ambiguous triples can still be oriented correctly under a different set of assumptions (e.g., V-OUS and collider-stability). Unfortunately, the authors didn't consider the mentioned work (except in the appendix briefly).

2. Here comes the main weakness. The SEM simulations in Section 4 are highly unclear. How many causal variables were considered? Was it a single number or a range? (I suspect that they are all small models with less than 20 variables.) How dense were the simulated graphs? How were the linear coefficients and noise terms sampled and from what distributions? Were the 100 tests sampled from the same configuration? Did V-OUS, collider-stability, and SMR hold in those simulations? Your readers can hardly reproduce your results without more details.

3. The comparison with the GRaSP algorithm is unfair in Example 2. While SMR fails in the example, GRaSP is not expected to give a correct answer (even asymptotically). With that said, it does not imply that Me-LoNS is a better algorithm. GRaSP/SP can identify the sparsest DAG while V-OUS and/or collider-stability fails (*). However, the authors seem to use Table 2 to imply that one algorithm is better than the other. Careful clarification is needed to avoid comparing apples to oranges (just as how the authors have remarked in Figure 4 that Me-LoNS and SP are different). The authors are advised to construct an example of (*) in the appendix to make this point clear.

4. Example 2 has another problem. The authors assumed that the true DAG is G0 so that they could compare Me-LoNS outputs with G0. However, what if the true DAG were G' instead? In that case, Me-LoNS would have an accuracy of 1-0.94=0.06 but 1-0.56=0.44 for GRaSP. Thus, the GRaSP statistic manages to reflect an inferential uncertainty that Me-LoNS cannot capture.

5. A "practical" comparison of Me-LoNS and GRaSP is missing (if the authors decided to compare the two). The latter aims at (1) handling almost-violations of faithfulness (particularly adjacency-faithfulness in their examples) in finite samples, and (2) an efficient greedy optimal search for large causal models. However, no practical efficiency result of Me-LonS is provided in this work. Indeed, consider the computational complexity of rules 1 and 2 in Definition 10. They need to exhaust all possible subsets C to determine whether a triple should be assigned as a collider or not. I can imagine how hard it is for Me-LoNS to handle a causal model with more than 50 variables.

**Q5 Detailed Comments To The Authors:**

In addition to the main weaknesses mentioned above, here are some minor comments.

(P.3) For orientation faithfulness, cite Ramsey et al. (2006) “Adjacency-Faithfulness and Conservative Causal Inference”. This is where the term originally came from.

(P.3) Capitalize OUS and ODS.

(P.4) Define “exchangeable distributions” (either in the main text or in the appendix).

(P.6) I think you mean G’ differs from G0 by flipping the edge X2 ~ X4 instead of X3 ~ X4.

(P.8) In Figure 4, claiming that “Me-LoNs” is different from (the consistency assumption required by) the SP algorithm is too vague. I believe you mean that they are logically independent (i.e., neither is stronger than the other).

(Other) Regarding my first comment in the main weakness section, the authors may want to consider a conservative version of Me-LoNS as well. For example, in line 3 of Me-LoNS, if there are DAGs from different MECs that satisfy the assignment of v-configurations, one may want to know of all possible ways to orient them (i.e., marking unshielded triples as ambiguous when modified v-stability fails) and a general method to represent uncertain orientations graphically.

**Q9 Complying With Reviewing Instructions:**

Yes

---

> ### Author Rebuttal · Authors · 2024-04-04
>
> Thank you for the detailed and insightful feedback, please find the following response to your comments:
>
> Q4
>
> 1. Yes, this a generalisation of Ramsey et al. (2006), in the sense that the conditions we propose are strictly weaker than the restricted faithfulness in Ramsey et al. (2006). We will include more references to Ramsey et. al (2006) in the final revision.
>
> 2. In our examples we consider 4 causal variables, $X_{1,2,3,4}$  generated via the SEMs 1 and 2 respectively on page 6. SEM 1 generates the noise as i.i.d. normal and SEM 2 generates the noise as i.i.d. Bernoullis, both using the scipy.stats package.
> In each comparison test with PC, we run both PC and Me-LoNS on the same input data of 10000 samples/data points sampled from SEM 1, we then repeat this test 100 times using differently sampled 10000 data points from SEM 1, and record the percentage of tests that the algorithms are consistent. Likewise for the SP comparison, but with SEM 2 instead. The simulations are meant to highlight the cases where Me-LoNS recovers the correct graph over other algorithms, thus V-OUS and collider-stable holds for SEM 1 and SEM 2 and SMR does not hold in SEM 2. While running the simulations, we have also conducted some tests where the output of Me-LoNS coincides with PC for some benchmark datasets.
>
> 3. Yes, Example 2, as well as remarks like ‘Example 2 shows that Me-LoNS is a viable alternative to all greedy variants of SP since Me-LoNS works under different conditions’ on page 6, show that there are cases that GRaSP/SP works whereas Me-LoNS doesn’t. One such example is mentioned in Example 2 “Consider the example in the proof of Theorem 2.4 in Raskutti and Uhler [2018] in which SMR holds but adjacency faithfulness is violated.” We will explicitly include this example the final revision. Table 2 is meant to illustrate one of the examples (Example 2) when Me-LoNS works better than GRaSP.
>
> 4. In the case when G’ is the true DAG, then wrt the same P, both conditions (SMR and V-OUS+collider-stable) would fail, and Me-LoNS would now give a wrong output and GRaSP still can’t differentiate between the correct and wrong output; it is “guessing.”  The edge X2 – X4 also can’t be uncertain since the resulting MECs would change if the edge is flipped. In light of Remark 10 of our paper, consider the case when we have n independent components of such G’, then both Me-LoNS and GRaSP would return the correct output about 0% of the time if n is large enough (½^n).
>
> 5. Yes, GRaSP is meant to be a speed up of SP at the cost of stronger conditions than SP, and thus generally should scale much better than Me-LoNS, however here the focus of the work is meant to relax conditions while GRaSP focuses on speeding up the SP algorithm at the cost of strictly stronger conditions (Lam et. al 2022). See also Remark 5 on page 5 addressing some practical aspects of the proposed algorithm.
>
> Q5
>
> P.3-P.6 Thanks for alerting us to these minor mistakes and typos. We will update those in the final revision.
>
> P.8 Yes formally that would be the most precise, ‘is different from’ is used to make a quick point here in Figure 4, we will reword this in the final revision.

---

### Official Review · Reviewer_HgCh · 2024-03-20

**Q2-1 Originality-Novelty:** 1
**Q2-2 Correctness-Technical Quality:** 3
**Q2-5 Clarity Of Writing:** 4

**Q10 Ethical Concerns:**

No ethical concerns

**Q1 Summary And Contributions:**

This paper aims to relax the orientation faithfulness assumption for constraint-based causal discovery by using the natural structure learning framework. They introduce Me-LoNS that locally test if (non-)colliders are correct. A theoretical justification is given to show how the orientation faithfulness assumption can be more relaxed for natural structure learning algorithms and how Me-LoNS differs from SP. They show empirically that Me-LoNS performs better than PC, but the comparison measure is not well interpretable. Me-LoNS seem to be similar to conservative PC (CPC) (Ramsey et al., 2006), although there is no comparison.

Ramsey, Joseph, Peter Spirtes, and Jiji Zhang. "Adjacency-faithfulness and conservative causal inference." Proceedings of the Twenty-Second Conference on Uncertainty in Artificial Intelligence. 2006.

**Q2-3 Extent To Which Claims Are Supported By Evidence:**

2: Fair: the main claims are somewhat supported by evidence (but the experimental evaluation may be weak, or does not match entirely with the claims, important baselines may be missing, proofs contain important ideas but lack rigor, algorithmic details are only discussed superficially, references are imprecise, assumptions are not sufficiently motivated or explicated, etc.).

**Q2-4 Reproducibility:**

3: Good: key resources (e.g. proofs, code, data) are available and key details (e.g. proofs, experimental setup) are sufficiently well-described for competent researchers to confidently reproduce the main results.

**Q3 Main Strengths:**

- The paper is transparent and mathematically precise in which assumptions are necessary. Although there is no theoretical comparison to CPC, Me-LoNS could maybe be a generalized version of CPC as the rules are defined for natural structural learning algorithms, and a more precise version of the underlying assumptions of CPC (although, CPC only needs causal Markov and Adjacency-Faithfulness assumptions according to Ramsey et al. (2006) which makes CPC maybe less restrive than Me-LoNS).
- The paper includes an augmented comparison to the SP algorithm.
- Organized paper: well-structured which makes it easier to read along the introduced definitions.
- The paper includes the proofs and the code.

**Q4 Main Weakness:**

1. As far as I know the assumption that there are no unshielded colliders (Remark 1) is not well-known, and it is not justified. In Example 2 / Figure 3 there are shielded colliders in the graph which seems to conflict with the assumption in Remark 1.
2. Page 3: It is not clear why “Likewise, collider-stable is implied by ods and can be seen as a relaxation."
3. Me-LoNS seems similar to CPC, but there is no theoretical/empirical comparison. Page 3: Definition 10 seems similar to the conservative orientation rule for colliders and non-colliders of CPC. Also for Remark 5: CPC should be a form of Me-LoNS. I believe that it would be valuable to add results on how Me-LoNS performs compared to CPC (under Oracle adjacencies for both methods to make it a fair comparison) in terms of orientation precision/recall and computational complexity. As far as I understand, besides Remark 1 which is not assumed by CPC, the assumptions for CPC and Me-LoNS are similar, but I am curious to see what the opinions of the authors are on this. Moreover, in the Conclusion section, it is written that “Exponential run time which is comparable to the skeleton building step of SGS.”. This suggests that CPC should be more computationally efficient than Me-LoNS because CPC is more efficient than SGS by using some smart ways to select the possible separating sets for the conservative orientation rules.
4. Table 1: it is unclear how the consistent output is defined. It would be insightful to have orientation precision and recall (and use oracle adjacencies as input) as measures to compare Me-LoNS to other methods.
5.  Page 6: it is unclear what type of mistakes PC makes when running SEM as defined in equation (1). By running Fisherz tests (pcalg) on the simulated data as described by the SEM (1) it does not seem to be clear that PC will draw wrong conclusions due to orientation faithfulness violations but draws wrong conclusions due to adjacency faithfulness violations (which is not the focus of this paper if I am correct).

**Q5 Detailed Comments To The Authors:**

1. What are real-world situations causing orientation faithfulness violations as a motivation to relax this assumption?
3. Where in Sadeghi and Soo (2022) do they prove Proposition 1?
4. Page 3: paragraph “By Example 21 in … for SGS/PC [Raskutti and Uhler, 2018].” This paragraph is not so clear. Do you mean that Sadeghi and Soo (2022) show in Example 21 that conditions 1,2,3 in Proposition 1 are strictly weaker than the faithfulness assumption defined in Definition 2? How does condition 2 relate to Example 21?

7. Page 3: Definition 10 condition 2 seems more restrive than Definition 8 condition 1.
9.  Page 6: Could you comment on what type of mistakes PC makes when running SEM as defined in equation (1)? By running Fisherz tests (pcalg) on the simulated data as described by the SEM (1) it does not seem to be clear that PC will draw wrong conclusions due to orientation faithfulness violations but draws wrong conclusions due to adjacency faithfulness violations (which is not the focus of this paper if I am correct).
10. Maybe the following paper is also relevant to this work: Marx, Alexander, Arthur Gretton, and Joris M. Mooij. "A weaker faithfulness assumption based on triple interactions." Uncertainty in Artificial Intelligence. PMLR, 2021.

**Q9 Complying With Reviewing Instructions:**

Yes

---

> ### Author Rebuttal · Authors · 2024-04-04
>
> Thank you for your questions and pointing out useful and interesting references.
>
> Q4
>
> 1. In Remark 1, and in this paper, we are not assuming the absence of unshielded colliders, but are clarifying that we will call what some authors refer to as unshielded colliders as simply “colliders” throughout the paper. We indeed have no restricting assumptions on the existence of different types of colliders or non-colliders in the causal graph.
>
> 2. We are just referring to an analogy where in the same manner that V-OUS is implied by ous, collider-stability is implied by ods. Indeed for collider $i\rightarrow k\leftarrow j$, by adjacency faithfulness, $i\not\sim j$ implies $i\perp j| C$ for some $C$ without k, but via ods, we can remove the $k$ from $C$ to obtain a $C'$ without $k$.
>
> 3. Me-LoNS is different to CPC in the rule that assign colliders, but is similar to CPC in how unassigned/ambiguous v-configurations are allowed. The assumptions for Me-LoNS are strictly weaker than the restricted faithfulness assumption of PC, see the following comment on page 4: “Thus the conditions in Theorem 1 is strictly weaker than those in Proposition 1, which is already weaker than restricted faithfulness”. It is true that CPC is more efficient run time wise, but the goal of the work here is mainly to weaken the assumptions under which the structure-learning algorithm works. We will highlight these comparisons to CPC in the final revision.
>
> 4. Informally, by consistent, we mean that the output of the algorithm is correct. Early in the paper, on page 2, we say that an algorithm is consistent when it outputs a graph correct up to MEC. We will mention the definition again in Section 4 in the final revision.
>
> 5. PC makes a mistake in classifying the colliders the graph of Figure 2 as non-colliders, when performing the orientation rules, summarised by the following comment on page 6: ``PC fails to identify the colliders in Figure 2”. Adjacency faithfulness violations are indeed not the focus and the error in PC is mostly due to orientation faithfulness violations in identifying/orienting colliders.
>
> Q5
>
> 1. Consider section 3.1.1 in our paper which discussed a mechanism of how orientation faithfulness can be violated (thus giving rise to our V-OUS condition), via conditional exchangeability on non-collider random variables (commonly made when random variables are indistinguishable from each other), and composition assumptions (satisfied by commonly used distributions like Gaussians). See also the comment “The V-OUS assumption can be interpreted as a prevention of Simpson’s paradox.” in Section 3.1.1 when our V-OUS condition can arise if Simpson’s Paradox doesn’t happen. Generally, for faithfulness, the reference ``Uhler et. al 2013” in our paper has shown that on a finite sample, empirical faithfulness violations can appear surprisingly often.
>
> 2. Remark 3 of our paper mentions that the converse pairwise Markovian condition for Theorem 25 of Sadeghi and Soo (2022) is, because of Theorem 14 of Sadeghi and Soo (2022), only really used to show adjacency faithfulness. Proposition 1 is thus obtained by replacing the converse pairwise Markovian condition in Theorem 25 of Sadeghi and Soo (2022) with adjacency faithfulness.
>
> 3. Yes, via Example 21 in Sadeghi and Soo (2022) conditions 1,2,3 combined are weaker than the faithfulness assumption in Definition 2. Condition 2 in Proposition 1 (ous+ods) holds for Example 21 in Sadeghi and Soo (2022), while the faithfulness assumption does not hold for Example 21 in Sadeghi and Soo (2022).
>
> 4. Definition 8 is a condition, whereas Definition 10 is an orientation rule; rule 1 and 2 are obtained from negating the corresponding conditions, 1 and 2 in Definition 8.
>
> 5. PC mistakenly classify the colliders the graph of Figure 2 as non-colliders, when performing the orientation rules, as remarked by the following comment on page 6: “PC fails to identify the colliders in Figure 2”. The error in PC is mostly due to orientation faithfulness violations in identifying/orienting colliders. Indeed, adjacency faithfulness is not the focus here.
>
> 6. Thank you for pointing out the UAI 2021 paper.  It turns out that the notion of 2-adjacency faithfulness and 2-orientation faithfulness reduces to regular adjacency and orientation faithfulness under certain circumstances, which are easy to verify when only singletons are involved. Example 2 in our paper does not satisfy the conditions described in the UAI 2021 paper, but does not pose a problem for our theory.  We did not weaken the common assumption of adjacency faithfulness, but rather focused on relaxing the orientation faithfulness conditions; thus providing necessary and sufficient conditions for our algorithm. However, both our papers coincide on a high level, by weakening existing conditions and providing a corresponding algorithm.

---

### Official Review · Reviewer_EZZt · 2024-03-21

**Q2-1 Originality-Novelty:** 2
**Q2-2 Correctness-Technical Quality:** 3
**Q2-5 Clarity Of Writing:** 3

**Q1 Summary And Contributions:**

The paper introduces a new class of structure learning algorithms that are guaranteed to return the correct CPDAG under weaker assumptions than faithfulness. The results extend those by Sadeghi and Soo (2022), who introduced natural structure learning algorithms, relaxing the faithfulness assumption, and this paper relaxes the assumptions in Sadeghi and Soo (2022). In addition, the authors introduce an algorithm of this type (Me-LoNS), which they compare to other existing algorithms.

**Q2-3 Extent To Which Claims Are Supported By Evidence:**

3: Good: the main claims are supported by convincing evidence (in the form of adequate experimental evaluation, proofs, (pseudo-)code, references, assumptions).

**Q2-4 Reproducibility:**

3: Good: key resources (e.g. proofs, code, data) are available and key details (e.g. proofs, experimental setup) are sufficiently well-described for competent researchers to confidently reproduce the main results.

**Q3 Main Strengths:**

The paper presents new results that allow one to relax the faithfulness assumption, and it appears to be technically sound.

**Q4 Main Weakness:**

(1) Given that this is an extension of Sadeghi and Soo (2022), I'm not sure how much of a novel contribution it is.

(2) The simulation study only considers two specific setups. I would have liked to see a more nuanced comparison, e.g. of aspects where the Me-LoNS does not outperform other methods.

**Q5 Detailed Comments To The Authors:**

(1) In light of Q4, I would have liked the authors to discuss how much of an improvement of the results from Sadeghi and Soo (2022) this paper presents. How much do they weaken the assumptions? Is the main advantage an improved run time?

(2) The authors somewhere refer to the condition sk(P) = sk(G), while they refer to adjacency faithfulness somewhere else. I believe that the two are equivalent (c.f. Proposition 9), so this creates unnecessary confusion. In addition, they write "Then if P is adjacency faithful wrt G, sk(P) = sk(G)" just before defining adjacency faithfulness (Definition 5), which is also confusing.

(3) The writing could be improved. The authors sometimes make grammatical errors: E.g., they sometimes mix up "have" with "has" and "are" with "is".

**Q9 Complying With Reviewing Instructions:**

Yes

---

> ### Author Rebuttal · Authors · 2024-04-04
>
> Thank you for giving helpful feedback and suggestions.
>
> Q4
>
> 1. Sadeghi and Soo (2022) described properties and conditions for natural structure learning algorithms to work, without providing a concrete algorithm.  Here, we provide one such algorithm Me-LoNS, while further weakening the described conditions.
>
> 2. One such example is mentioned in Example 2 on Page 6, where we ``consider the example in the proof of Theorem 2.4 in Raskutti and Uhler [2018] in which SMR holds but adjacency faithfulness is violated.’’  We will be sure to include this example, in the final revision.
>
> Q5
>
> 1. No actual computer implementable algorithm is provided by Sadeghi and Soo (2022).  Here we provide one such runnable algorithm. The assumptions are weakened from a global order stability condition (ous and ods) to a *local* V-OUS+collider-stable condition.
>
> 2. The condition $sk(P)=sk(G)$ is meant to highlight the algorithmic nature of how we handle adjacency faithfulness in this work, however we will make the notation more consistent in the final revision.
>
> 3. We have identified a couple of such errors and will carefully correct them in the final revision.

---

### Official Review · Reviewer_2HB2 · 2024-03-23

**Q2-1 Originality-Novelty:** 3
**Q2-2 Correctness-Technical Quality:** 3
**Q2-5 Clarity Of Writing:** 4

**Q1 Summary And Contributions:**

The authors extend the class of natural structure learning algorithms by presenting a strict relaxation of the faithfulness assumption. The proposed Modified V-stable Localized Natural Structure learning algorithm (MeLoNS) could be seen as a generalization of the PC algorithm in terms of how v-structure orientation rules are determined. MeLoNS has an exponential run-time.

**Q2-3 Extent To Which Claims Are Supported By Evidence:**

3: Good: the main claims are supported by convincing evidence (in the form of adequate experimental evaluation, proofs, (pseudo-)code, references, assumptions).

**Q2-4 Reproducibility:**

3: Good: key resources (e.g. proofs, code, data) are available and key details (e.g. proofs, experimental setup) are sufficiently well-described for competent researchers to confidently reproduce the main results.

**Q3 Main Strengths:**

* The proof accompany their algorithm with rigorous theoretical proofs along with intuitions in order to help understand the theory better.
* Even though it might seem like an incremental work, I feel the proposed algorithm with the relaxation is quite strong
* The authors provide experiments on consistency of the algorithm showing that the proposed algorthm is consistent compared to other approaches.
* Well written and well structured paper.

**Q4 Main Weakness:**

* Counter examples to the proposed algorithm with conditions and examples where the algorithm might fail are not presented.

**Q5 Detailed Comments To The Authors:**

The work compares SP and PC with MeLoNS and provides examples of SEMs which the existing techniques fail to discover. In the case of PC this would the cases where faithfulness is violated. Can a similar example be provided for MeLoNS? Or in other words, under what conditions over the ground truth SEM, could the algorithm fail (assuming the conditional independence oracle)?

Faithfulness gives us two implications of Adjacency Faithfulness and Orientation Faithfulness with PC checking for the latter. If orientation faithfulness fails, PC will fail as well. I am having trouble understanding how does the proposed algorithm relate to this notion. Am i correct in thinking that the borrowed part from PC i.e. the skeleton discovery will only describe the edges for the DAG. And MeLoNS replaces the alignment of unshielded triples and the Meek's rules steps - So the Orientation and Adjacency faithfulness can both be relaxed to the proposed conditions. It will be good to hear some comments on this.

**Q9 Complying With Reviewing Instructions:**

Yes

---

> ### Author Rebuttal · Authors · 2024-04-04
>
> Thank you for your helpful comments and suggestions.
>
> Q4
> Note that the given conditions in Theorem 1 are both sufficient and necessary.  Thus, we know precisely when the algorithm fails, and one such example is mentioned in Example 2 on page 6, where we ``consider the example in the proof of Theorem 2.4 in Raskutti and Uhler [2018] in which SMR holds but adjacency faithfulness is violated.’’   We will explicitly include this example in the final revision.
>
> Q5
> As mentioned in Q4, the quoted example from Raskutti and Uhler [2018] in Example 2, would serve as the counter-example.
>
> Yes, in Me-LoNS the orientation and Meek’s rule part of PC is replaced, while the skeleton discovery part is the same.  Adjacency faithfulness, which is related to the skeleton discovery part, is not relaxed, but Me-LoNS does relax orientation faithfulness, via the new orientation steps.

---

### Meta-Review · Area_Chair_dCiz · 2024-04-17

This paper defines a localized version of natural structure learning algorithms by relaxing the consistency conditions. Also, a practical algorithm is proposed under these conditions. Simulation studies show that the proposed method outperforms traditional PC/SGS algorithms. I would recommend accepting this paper given its novelty and technical contribution. However, the authors should carefully revise the paper according to the suggestions from all reviewers. In particular, the discussion of difference to CPC and experimental comparison to CPC needs to be added.